# REMA: A Unified Reasoning Manifold Framework for Interpreting Large Language Model

## Abstract

Understanding how Large Language Models (LLMs) perform complex reasoning and their failure mechanisms is a challenge in interpretability research. To provide a measurable geometric analysis perspective, we define the concept of the "**Reasoning Manifold**", a latent low-dimensional geometric structure formed by the internal representations corresponding to all correctly reasoned generations. This structure can be conceptualized as the embodiment of the effective thinking paths that the model has learned to successfully solve a given task. Based on this concept, we build **REMA**, a framework that explains the origins of failures by quantitatively comparing the spatial relationships of internal model representations corresponding to both erroneous and correct reasoning samples. Specifically, REMA first quantifies the geometric deviation of each erroneous representation by calculating its $k$-nearest neighbors distance to the approximated manifold formed by correct representations, thereby providing a unified failure signal. It then localizes the divergence points where these deviations first become significant by tracking this deviation metric across the model's layers and comparing it against a baseline of internal fluctuations from correct representations, thus identifying where the reasoning chain begins to go off-track. Our extensive experiments on diverse LLMs and multimodal LLMs and tasks demonstrate the low-dimensional nature of the reasoning manifold and the high separability between erroneous and correct reasoning representations. The results also validate the effectiveness of the REMA framework in analyzing the origins of reasoning failures. This research connects abstract reasoning failures to measurable geometric deviations in representations, providing new avenues for in-depth understanding and diagnosis of the internal computational processes of black-box models. Our code will be publicly available when the paper is accepted.

## 1 Introduction

Understanding how Large Language Models (LLMs) perform complex reasoning and their failure mechanisms is a challenge in interpretability research. Fefferman et al. (2016); Tan et al. (2024); Song & Zhong (2023) have revealed the highly structured and static nature of the internal activation space. On the other hand, mechanistic interpretability (Murdoch et al., 2019), through techniques such as circuit analysis (Meng et al., 2023; Palit et al., 2023) and causal tracing (Palit et al., 2023), has successfully deconstructed how models implement known functions. In addition, analysis of failure modes such as hallucination (Huang et al., 2025) or hierarchical failures (Ko et al., 2024), while focused on errors, often rely on signals specific to an error type or controlled input contrasts. In parallel work, Curved Inference (Manson, 2025), treats reasoning as a geometric trajectory in the residual stream and uses metrics such as curvature and salience to analyze how trajectories bend in response to semantic concerns in the input. However, these methods often rely on probes designed for specific error types or on analyzing representation changes by contrasting controlled input pairs. A general framework that can localize diverse, naturally occurring reasoning failures from the geometric structure of internal representations is still lacking.

To establish a measurable, geometric analysis perspective, we introduce the Reasoning Manifold as the theoretical cornerstone of our framework. We posit that when an LLM learns an effective reasoning strategy for a task, the internal representation sequences of its correct inference processes do not scatter randomly throughout the high-dimensional activation space. Instead, these successful

thinking trajectories tend to concentrate and evolve on a relatively low-dimensional, structured subspace. We define this geometric structure formed by correct reasoning paths in the representation space as the Reasoning Manifold. This concept is a natural extension of the widely recognized manifold hypothesis from machine learning (Fefferman et al., 2016; Narayanan et al., 2010; Bengio et al., 2013) to the process of LLM reasoning, which allows us to translate the abstract notion of correct thought into an analyzable geometric object. Building upon this core concept of the Reasoning Manifold, we propose **REMA**, a novel post-hoc framework designed to diagnose the origins of failures by analyzing the geometry of the representation space. The central mechanism of REMA is to unify all reasoning failures as a geometric deviation of their internal representations from the correct reasoning manifold. To operationalize this idea, REMA follows a two-step analytical workflow: First, to quantify the severity of a failure, it computes the $k$-nearest neighbor distance of each erroneous representation to the approximated manifold formed by the correct representations, thereby providing a model-agnostic and unified deviation signal for each error. Second, to localize the origin of a failure, it identifies the divergence point by tracking the deviation distance of a single error sample layer-by-layer and statistically comparing it against a baseline of normal fluctuations within the correct representations. This divergence point precisely marks the layer where the model's reasoning chain begins to go off-track.

We validate the effectiveness of REMA through extensive experiments covering both LLMs and MLLMs on multiple reasoning tasks. Our main findings are as follows: (1) internal states of both correct and incorrect reasoning tend to exhibit low-dimensional structures, providing an empirical basis for manifold analysis; (2) representations of erroneous reasoning consistently show statistically significant geometric deviation from the manifold of correct reasoning; and (3) these deviations can be traced back to specific model layers, revealing model- and task-dependent patterns of failure origins.

The main contributions of this work can be summarized as follows. **(I)** We introduce and systematically investigate the concept of the Reasoning Manifold in LLMs, offering a new theoretical perspective on the structure of internal reasoning states. **(II)** We propose REMA, a post hoc interpretability framework that leverages geometric analysis to quantitatively analyze the representation deviation associated with incorrect reasoning and to locate the origins of the failure. **(III)** Through experiments on multiple models and tasks, we provide strong evidence for the geometric deviation hypothesis of reasoning failures and demonstrate the value and generality of REMA in diagnosing these failures and their originating stages.

## 2 RELATED WORK

Our research aims to understand the reasoning processes of Large Language Models (LLMs) by analyzing the geometric structure of their internal representations. This section reviews relevant state-of-the-art research, progressively revealing the limitations in current understanding and thereby situating the unique contributions of our REMA framework.

Modeling the activation space of LLMs as a geometric object is a powerful pathway to understanding their functionality. The Manifold Hypothesis (Fefferman et al., 2016; Narayanan et al., 2010; Bengio et al., 2013) posits that meaningful data variations are often confined to low-dimensional manifolds embedded within a high-dimensional space. Recent studies have confirmed that despite the massive scale of model parameters, the Intrinsic Dimension (ID) of their activations during specific tasks can be significantly lower (Facco et al., 2017; Pope et al., 2021). The analysis of representation entropy by Tan et al. (2024) further connects information theory with LLM geometry, exploring scaling laws by relating entropy to model size. Song & Zhong (2023) proposed a linear decomposition of hidden states into interpretable geometric components such as position and context, analyzing the properties of these fundamental building blocks.

Mechanistic Interpretability (Murdoch et al., 2019) is dedicated to explaining how these structures support the model's reasoning processes. Early work focused on analyzing attention mechanisms (Vaswani et al., 2023; Cao et al., 2020) to understand model focus. Recent work has advanced through probing tasks to assess the information encoded in representations (Tenney et al., 2019; Hao et al., 2021; Salin et al., 2022; Liu et al., 2024; Zhang et al., 2024; Liu et al., 2023b; Beigi et al., 2024), or by associating neural activations with human-understandable concepts via methods like representation disentanglement (Bau et al., 2017; Gandelsman et al., 2024; Balasubramanian et al., 2024). Sparse

Auto Encoders (SAEs) have been proposed as a powerful technique to decompose model activations into sparse and more monosemantic features (Yun et al., 2021; Cunningham et al., 2023; Daujotas, 2024). Deeper causal inference techniques, such as activation patching or circuit analysis (Meng et al., 2023; Palit et al., 2023; Basu et al., 2024; Yu & Ananiadou, 2024; Conmy et al., 2023; Syed et al., 2023), attempt to locate and understand the internal computational mechanisms that implement specific functions.

However, understanding how a model works correctly is insufficient; we critically need to know why and how it fails. An important direction in failure analysis is the study of hallucination and factual errors, where research suggests that signals of deviation may be detectable even in the early layers of a model (Huang et al., 2025). Recent work has also proposed a hierarchical failure framework, revealing that reasoning errors may originate from different cognitive stages, such as an initial number recognition stage versus a later arithmetic logic stage (Ko et al., 2024; Song et al., 2025). Concurrent work, the Curved Inference framework (Manson, 2025), treats reasoning as a geometric trajectory in the residual stream and uses metrics like curvature and salience to analyze how trajectories bend in response to "semantic concerns" in the input.

Nevertheless, these methods often rely on probes designed for specific error types (e.g., hallucinations) or on analyzing representation changes by contrasting controlled input pairs (e.g., neutral vs. emotional prompts). The innovation of REMA lies in proposing a unified, task-agnostic definition of failure: all types of reasoning errors (be they logical, factual, or otherwise) can be attributed to their internal representations deviating from the manifold formed by naturally occurring correct reasoning samples from a benchmark task. We focus on the outcome of reasoning (success vs. failure), not just semantic shifts in the input. Our core metric—deviation distance—measures the position of a representation relative to a normative population, rather than the intrinsic shape of the trajectory itself (like curvature).

## 3 METHODOLOGY: THE REMA FRAMEWORK

In this section, we introduce the REMA framework. REMA is designed to interpret the reasoning processes of LLMs and localize their origins of failure by analyzing the geometric structure of their internal representations. We will sequentially introduce problem formulation (Section 3.1), the core hypothesis of the Reasoning Manifold (Section 3.2), and the two core mechanisms for analyzing reasoning failures: deviation and separability analysis (Section 3.3) and the localization of reasoning divergence points (Section 3.4).

### 3.1 PROBLEM FORMULATION AND REPRESENTATION EXTRACTION

Let $\mathcal{D} = \{(X_i, Y_i^*)\}_{i=1}^N$ be a reasoning dataset of $N$ samples, where $X_i$ represents the $i$-th sample's input (which can be text-only, or a multimodal pair such as image $X_{v,i}$ and text question $X_{t,i}$), and $Y_i^*$ is the corresponding ground truth answer. We use a pre-trained LLM $M$ to perform zero-shot inference on $X_i$, obtaining the model's predicted output sequence $Y_i = (y_{i,1}, ..., y_{i,T_i}) = M(X_i)$.

To facilitate our representation analysis, we partition the samples by comparing the final textual content of the model's prediction $Y_i$ with the ground truth $Y_i^*$. While some tasks may involve more complex evaluation metrics, we adopt strict exact matching (case-insensitive, ignoring leading/trailing whitespace) as a uniform and unambiguous criterion for partitioning across diverse tasks. This partitions the samples into two subsets: the correct reasoning set $\mathcal{D}_{\text{correct}}$ and the error reasoning set $\mathcal{D}_{\text{error}}$. Since complex semantic understanding and logical inference are primarily handled by the model's core language module (typically a decoder) (Liu et al., 2023a; Li et al., 2023), our study focuses on analyzing the hidden state representations within this module. Let $L$ be the total number of layers in this module. During the auto-regressive generation of the output sequence $Y_i$ at each step $t$, we extract the hidden state vector from the position of the last token of each computational layer $l$ (from 0 to $L-1$), which is used to predict the next token $y_{i,t}$. This state is denoted as $\mathbf{z}_{i,t}^l \in \mathbb{R}^d$. To obtain an aggregated representation for each sample in a specific layer $l$, we calculate the mean distributed vector of its hidden states $\mathbf{z}_{i,t}^l$ in all $T_i$ generation steps.

$$\mathbf{z}_i^l = \frac{1}{T_i} \sum_{t=1}^{T_i} \mathbf{z}_{i,t}^l \qquad (1)$$

Mean pooling is a common technique for summarizing variable-length sequential information into a fixed-dimensional vector (Cer et al., 2018; Devlin et al., 2019; Reimers & Gurevych, 2019), providing a generalized representation of the layer's average activation state. We collect these mean-pooled representations for correct and error samples at each layer $l$: $\mathcal{Z}_{\text{correct}}^l = \{\mathbf{z}_i^l | (X_i, Y_i) \in \mathcal{D}_{\text{correct}}\}$ and $\mathcal{Z}_{\text{error}}^l = \{\mathbf{z}_i^l | (X_i, Y_i) \in \mathcal{D}_{\text{error}}\}$.

## 3.2 REASONING MANIFOLD

Our methodology is built upon the Reasoning Manifold Hypothesis, which is inspired by the widely-recognized Manifold Hypothesis in machine learning (Fefferman et al., 2016; Narayanan et al., 2010; Bengio et al., 2013). We hypothesize that for a given LLM $M$ and a specific reasoning task $\mathcal{D}$, there exists a latent, typically low-dimensional Reasoning Manifold $\mathcal{R}^l$, embedded in the high-dimensional representation space $\mathbb{R}^d$ at layer $l$. This manifold $\mathcal{R}^l$ is primarily constituted by or closely enveloped by the representations of correctly reasoned samples, $\mathcal{Z}_{\text{correct}}^l$. In other words, when the model successfully performs reasoning, its internal states tend to evolve and reside in these structured, lower-dimensional subspaces. We further hypothesize that reasoning failures are often associated with the model's internal representation states significantly deviating from these learned manifolds for correct reasoning. Since the true manifold $\mathcal{R}^l$ is unknown, we use the point cloud of correct reasoning representations $\mathcal{Z}_{\text{correct}}^l$ as its empirical approximation, denoted $\hat{\mathcal{R}}^l$. We characterize the properties of $\hat{\mathcal{R}}^l$ using Intrinsic Dimension (ID) and Mutual Information (MI) (see Appendix A) and compare them with the corresponding properties of the error representation set $\mathcal{Z}_{\text{error}}^l$.

## 3.3 DEVIATION AND SEPARABILITY ANALYSIS

We validate our core hypothesis by quantifying the deviation of error reasoning representations $\mathcal{Z}_{\text{error}}^l$ from the approximated correct reasoning manifold $\hat{\mathcal{R}}^l$ (represented by $\mathcal{Z}_{\text{correct}}^l$), and by testing the separability of these two sets of representations.

**Deviation Distance.** For each error representation $\mathbf{z}_j \in \mathcal{Z}_{\text{error}}^l$, we compute its deviation $D_j^l$ as the average Euclidean distance to its $k'$ nearest neighbors in the correct representation set $\mathcal{Z}_{\text{correct}}^l$. The average deviation distance for all error samples is $D_{\text{error}}^l = \text{mean}_j(D_j^l)$. As a baseline, for each correct representation $\mathbf{z}_i \in \mathcal{Z}_{\text{correct}}^l$, we calculate its average Euclidean distance $d_i^l$ to its $k'$ nearest neighbors within $\mathcal{Z}_{\text{correct}}^l$ (excluding itself). The average internal distance for correct samples is $D_{\text{correct}}^l = \text{mean}_i(d_i^l)$. We use a $k'$-nearest-neighbor algorithm to compute these distances. To determine whether the deviation is significant, we perform a two-sample Welch t-test on the distributions of $\{D_j^l\}_j$ and $\{d_i^l\}_i$.

**Separability Test.** To further assess whether correct and error reasoning representations are spatially distinguishable, we train a binary classifier for each layer $l$ on the representation sets $\mathcal{Z}_{\text{correct}}^l$ (labeled as class 0) and $\mathcal{Z}_{\text{error}}^l$ (labeled as class 1). We use a Support Vector Classifier (SVC) with a Radial Basis Function (RBF) kernel. The features are a reality of standardization and the class weights are balanced to handle potential sample imbalance. The classifier's ability to distinguish between the two classes is evaluated using a 5-fold cross-validation, reporting the mean accuracy.

## 3.4 LOCALIZATION OF DIVERGENCE POINT

We aim to identify the layer at which an individual's incorrect reasoning path begins to deviate significantly from the correct reasoning region, termed the divergence point $l_{\text{diverge}}$. For each error sample $j \in \mathcal{D}_{\text{error}}$, we compute its deviation $D_j^l$ in each layer $l$ as defined in Section 3.3. We also use the mean $\mu_{\text{correct}}^l$ and standard deviation $\sigma_{\text{correct}}^l$ of the internal distances $\{d_i^l\}_i$ of the correct samples at layer $l$ (from Section 3.3). To formalize the definition of the divergence point, we set a statistics-based threshold. Specifically, if the deviation of an error sample at a certain layer exceeds the mean internal distance of correct samples plus multiple standard deviations, we regard this as the onset of a significant divergence:

$$D_j^l > \mu_{\text{correct}}^l + \alpha \cdot \sigma_{\text{correct}}^l \tag{2}$$

where $\alpha$ is a pre-set threshold factor (e.g., $\alpha = 2$ in this study). The divergence point $l_{\text{diverge},j}$ for error sample $j$ is then defined as the minimum (earliest) layer index $l$ that satisfies this condition. By

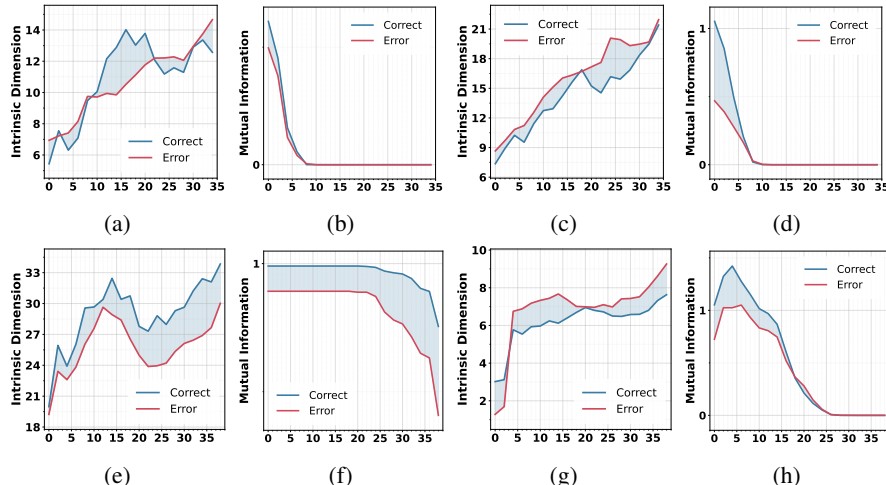

Figure 1: Layer-wise ID and MI of reasoning manifolds for four tasks, including Qwen3 (4B) on `MATH` (a-b) and `GSM8K` (c-d), and Llama3.2 (11B) on `SNLI-VE` (e-f) and `MathVista` (g-h). **BLUE** lines represent correct reasoning states and **RED** lines represent error reasoning states. The X-axis indicates layer id of LLMs. The result show that all ID are significantly lower than the original hidden space dimensionality, and the MI is relatively high in the early layers and gradually decreases with depth.

analyzing the distribution of $l_{\text{diverge},j}$ across all error samples, we can understand at which processing stage the model reasoning failures tend to originate.

## 4 EXPERIMENTS

In this section, we describe our experimental setup and evaluate the effectiveness of our proposed REMA framework across multiple LLMs and reasoning benchmarks.

### 4.1 EXPERIMENTAL SETUP

**Datasets.** To comprehensively assess the analytical capabilities of REMA across different reasoning scenarios, we carefully selected a total of seven benchmark datasets, covering both text reasoning and multimodal reasoning. The selection aims to encompass a diverse spectrum of reasoning abilities, from general visual question answering to complex multi-step mathematical deduction, specifically including text reasoning tasks: **GSM8K** (Cobbe et al., 2021) (arithmetic), **MATH** (Hendrycks et al., 2021) (math reasoning), **GPQA** (Rein et al., 2024) (scientific questions), and multimodal reasoning tasks: **VQAv2** (Goyal et al., 2017) (visual question answering), **SNLI-VE** (Xie et al., 2019) (visual comprehension task), **AI2D** (Kembhavi et al., 2016) (comprehension of diagrams and charts), **MathVista** (Lu et al., 2024) (visual mathematical reasoning). This diversity allows us to test the generalizability and effectiveness of the REMA framework across different reasoning scenarios. For all experiments, we used the official validation or test splits of these datasets. Further details and the accuracy resulting on each dataset are provided in Appendix B.

**Models.** To validate the effectiveness of REMA across different model architectures and parameter sizes, we selected a range of widely recognized and used by academia and industry, open-source LLMs and MLLMs. Our selection covers multiple model families and parameter sizes ranging from 3B to 90B. Specifically, they include: Qwen3 (Yang et al., 2025), Qwen2.5-VL (Bai et al., 2025), Llama3.2 (Grattafiori et al., 2024), and LLaVA-OneVision (Li et al., 2024). This diversity enables us to evaluate the analytical capabilities of the REMA framework across different model configurations and to explore the potential influence of model characteristics on the structure of reasoning manifolds, thereby covering a broader range of application scenarios. The model details, including parameter and base architectures, are provided in Appendix C.

Table 1: Average deviation distance summary across all layers for each model and task, integrated with model performance. "Rel. Dev." denotes the relative deviation distance, calculated as $(D_{\text{error}}/D_{\text{correct}}) - 1$. A Spearman's rank correlation of $\rho = 0.598$ ($p < 0.01$) was found between Accuracy and Relative Deviation across all model-task pairs.

| Model | Task | | Acc | Avg. Error Dist. | Avg. Correct Dist. | Rel. Dev. | T-stat |
|---|---|---|---|---|---|---|---|
| Llama3.2 (3B) | Text | GPQA | 20.6% | 5.03 | 3.76 | 0.34 | 12.41 |
| | | GSM8K | 10.3% | 3.89 | 3.08 | 0.26 | 10.87 |
| | | MATH | 11.6% | 4.40 | 3.45 | 0.28 | 7.59 |
| Qwen3 (4B) | Text | GPQA | 26.4% | 37.31 | 29.40 | 0.27 | 10.78 |
| | | GSM8K | 22.3% | 33.38 | 25.16 | 0.33 | 24.82 |
| | | MATH | 17.4% | 41.58 | 30.22 | 0.38 | 15.39 |
| LLaVA-OneVision (7B) | Text | GPQA | 23.4% | 38.45 | 31.18 | 0.23 | 9.82 |
| | | GSM8K | 14.6% | 42.83 | 34.96 | 0.22 | 13.98 |
| | | MATH | 13.6% | 48.47 | 39.95 | 0.21 | 7.32 |
| | Multimodal | AI2D | 72.0% | 39.69 | 29.24 | 0.36 | 33.66 |
| | | MathVista | 53.7% | 40.17 | 27.97 | 0.44 | 21.94 |
| | | SNLI-VE | 93.4% | 10.16 | 7.02 | 0.45 | 29.20 |
| | | VQAv2 | 71.8% | 54.22 | 33.45 | 0.62 | 39.05 |
| Qwen2.5-VL (3B) | Text | GPQA | 23.0% | 35.51 | 28.15 | 0.26 | 10.00 |
| | | GSM8K | 14.3% | 33.90 | 27.77 | 0.22 | 11.76 |
| | | MATH | 11.4% | 42.68 | 35.41 | 0.21 | 5.40 |
| | Multimodal | AI2D | 73.5% | 29.03 | 22.61 | 0.28 | 21.17 |
| | | MathVista | 55.8% | 33.97 | 21.53 | 0.53 | 22.81 |
| | | SNLI-VE | 97.8% | 5.50 | 3.70 | 0.49 | 18.25 |
| | | VQAv2 | 66.4% | 40.94 | 23.73 | 0.73 | 32.56 |
| Llama3.2 (11B) | Text | GPQA | 23.3% | 6.46 | 5.14 | 0.26 | 10.27 |
| | | GSM8K | 12.2% | 6.92 | 4.90 | 0.41 | 19.39 |
| | | MATH | 11.8% | 8.10 | 6.13 | 0.32 | 9.34 |
| | Multimodal | AI2D | 71.2% | 5.66 | 4.52 | 0.25 | 24.88 |
| | | MathVista | 36.9% | 7.06 | 4.85 | 0.46 | 23.56 |
| | | SNLI-VE | 76.6% | 1.45 | 0.91 | 0.59 | 58.93 |
| | | VQAv2 | 66.1% | 8.75 | 5.43 | 0.61 | 48.72 |

## 4.2 CHARACTERIZATION ANALYSIS

To validate the Reasoning Manifold hypothesis and characterize its properties, we computed the ID and task-relevant MI for the sets of correct and erroneous reasoning samples at each layer of the LLMs. Figure 1 illustrates the trends of these two metrics as a function of layer depth for Qwen3 (4B) on the MATH and GSM8K (top row), and for Llama3.2 (11B) on the SNLI-VE and MathVista (bottom row).

**ID.** As shown in the first and third columns of Figure 1, we observe that, regardless of whether the reasoning is correct or erroneous, all estimated intrinsic dimensions are significantly lower than the original hidden space dimensionality (e.g. 2048 or 4096). This provides strong evidence that the internal reasoning processes of LLMs, both successful and failed, tend to operate within low-dimensional subspaces. This validates the rationale for using manifold analysis.

**MI.** The second and fourth columns of Figure 1 display the mutual information between the layer representations and the final correct answer. We first observe that, for both successful and failed reasoning, the MI is relatively high in the early layers and gradually decreases with depth. Second, correct samples exhibit higher MI values than error samples in the early layers, indicating that the representations of correct reasoning paths contain more useful information pertinent to the correct answer from the initial stages.

Furthermore, we found that the relative magnitudes of ID and MI across layers do not follow a single universal trend across different model families or tasks. It suggests that each model and task may possess its own "reasoning fingerprint", a distinctive layer-wise profile that reflects how the model organizes its inference process, which provides a strong indication that they are geometrically separable.

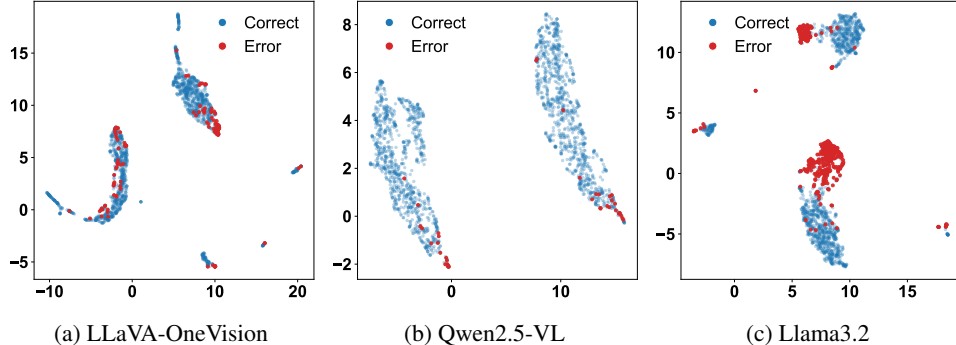

(a) LLaVA-OneVision       (b) Qwen2.5-VL       (c) Llama3.2

Figure 2: UMAP visualization of representations from the final layer on `SNLI-VE`. **BLUE** points represent correct reasoning samples and **RED** points represent error reasoning samples. The figures show a clear tendency for correct reasoning samples and error reasoning samples to separate.

### 4.3 DEVIATION ANALYSIS

In this section, we provide core evidence that the internal representations of erroneous reasoning deviate from the manifold region formed by correct reasoning. We validate this hypothesis through quantitative deviation distance analysis, UMAP visualization, and separability tests.

**(a) Quantitative Deviation Analysis.** To quantify the extent to which error representations deviate from the correct reasoning manifold, we computed the average $k'$-nearest neighbor distance of each error sample to the manifold and compared it with the average internal $k'$-nearest neighbor ($k = 5$) distance among the correct samples within the manifold. As shown in Table 1, across all evaluated tasks, the average deviation distance of error samples is consistently and significantly greater than the average internal distance of the correct samples. The average t-statistics are consistently high, indicating that this difference is highly statistically significant. This result provides strong support for our core hypothesis: when LLMs commit reasoning errors, their internal representations indeed deviate geometrically from the representational region formed by correct reasoning. Furthermore, we investigated the potential correlation between task difficulty and the magnitude of deviation. We use the model's zero-shot accuracy (Appendix C) for task difficulty (lower accuracy implies higher difficulty) and calculated the Spearman's rank correlation between accuracy and the relative deviation distance (defined as $(D_{\text{error}}/D_{\text{correct}}) - 1$). As illustrated in Table 1, we found a significant negative correlation between these two metrics (Spearman's $\rho = 0.598$, $p < 0.01$). This implies that on tasks where models perform worse (i.e., more difficult tasks), their error representations tend to deviate more severely from the correct representations. This may suggest that for more challenging tasks, the correct reasoning manifold itself is narrower or more sensitive to perturbations, causing any minor initial deviations to be amplified during multi-step reasoning, resulting in the final error representations being geometrically further from the correct region. Finally, despite these differences in absolute scale, the relative relationship where "error deviation is greater than the correct internal distance" is true across all models. This further demonstrates the general applicability of the REMA framework.

**(b) UMAP Visualization of Deviations.** To visually illustrate the distributional differences between correct and error reasoning representations in the latent space, we performed a UMAP dimensionality reduction on the hidden states from final layers. As depicted in Figure 2, for the `SNLI-VE` task, all models exhibit a clear tendency for correct reasoning samples and error reasoning samples to separate in the two-dimensional UMAP space at their final language model layer. Correct samples often form one or more relatively concentrated clusters, while error samples are either scattered at the periphery of these clusters or form distinct, separate smaller clusters. This visual separation provides intuitive evidence that error representations deviate from the regions of correct reasoning. To further validate the robustness of this finding and to investigate how this separation evolves layer-wise, we present additional visualizations using t-SNE in Appendix F.2, including a detailed analysis of a layer-by-layer t-SNE visualization for Llama3.2 (11B).

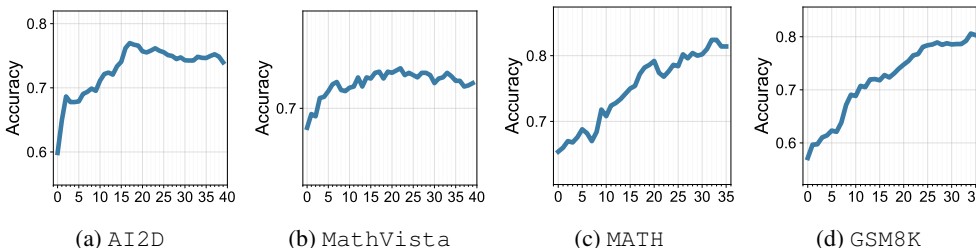

(a) `AI2D`    (b) `MathVista`    (c) `MATH`    (d) `GSM8K`

Figure 3: Layer-wise separability accuracy between correct and error reasoning representations for Llama3.2 (11B ) on `AI2D` and `MathVista` (a-b) and Qwen3 (4B) on `MATH` and `GSM8K` (c-d). The X-axis indicates layer id of LLMs. The accuracy typically increases progressively with layer depth, reaching a high and stable level in the mid-to-late layers of the model.

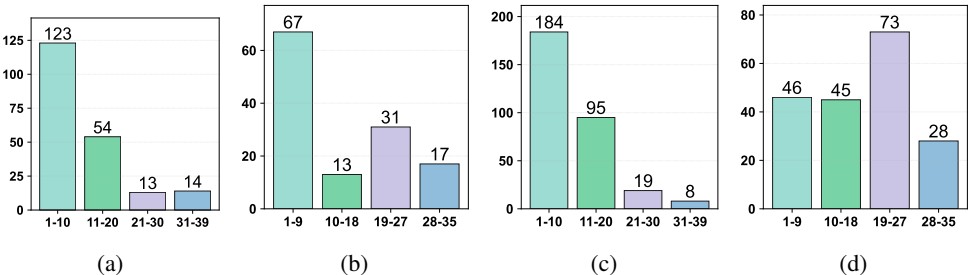

(a)    (b)    (c)    (d)

Figure 4: Distribution of reasoning divergence points across aggregated decoder layers for Llama3.2 (a-b) and Qwen3 (c-d) on the `AI2D` and `MathVista` . Each bar represents the count of error samples whose representations first significantly deviated from the correct reasoning manifold within an 8-layer interval. The X-axis indicates aggregated layer blocks. By sequentially tracking and thresholding the divergence of the hidden states of each error sample at different layers, we can pinpoint the specific point at which the internal representation starts to deviate significantly from the approximated correct reasoning trajectory.

**(c) Separability Test.** To further quantify, from a classifier's perspective, the distinguishability of correct versus error reasoning representations, we conducted a separability test on the representations from each layer. Figure 3 presents the layer-wise mean accuracy of an SVM classifier in distinguishing between correct and error representations for Llama3.2 (11B) and Qwen3 (4B). First, across all model-task combinations, the separability accuracy is generally far above random chance levels. Second, the accuracy typically increases progressively with layer depth, reaching a high and stable level in the mid-to-late layers of the model, sometimes approaching perfect separation. This indicates that the differences between correct and incorrect reasoning paths become increasingly pronounced as the layer depth increases, and our framework can effectively capture representational signals distinguishing correct from incorrect reasoning.

## 4.4 Localizing Reasoning Divergence

To gain a deeper understanding of the specific stages in which reasoning failures occur, we identified the minimum (earliest) layer index for each error sample where its internal representation began to deviate significantly from the approximate correct reasoning manifold. This layer is termed the divergence point. Figure 4 shows the histograms of the divergence point distributions for the four evaluated MLLMs on two representative datasets, `AI2D` (Row 1) and `MathVista` (Row 2). For clarity of visualization, we aggregated consecutive 8 decoder layers into a single bin. First, for the same task, divergence points of different models can occur at quite distant layers. For the same model, divergence points can also occur at quite distant layers for different tasks. These suggest that reasoning failures are not simply concentrated at a single, fixed stage of processing. Instead, they are highly dependent on the complex interplay between the model architecture and the task features. Second, Despite the high degree of heterogeneity in model behavior due to different datasets and architectures, the mechanism for localizing divergence points within our REMA framework shows its

Table 2: Ablation Study. Comparison of average deviation distances under different pooling strategies across models.

| Model | Llama3.2 | | | Qwen2.5-VL | | | Qwen3 | | |
|---|---|---|---|---|---|---|---|---|---|
| Task | SNLI-VE | | | SNLI-VE | | | GPQA | | |
| Pooling | Err Dist | Corr Dist | T-stat | Err Dist | Corr Dist | T-stat | Err Dist | Corr Dist | T-stat |
| w/ Last-token | 2.37 | 1.45 | 48.76 | 8.12 | 5.50 | 13.19 | 82.59 | 60.38 | 8.25 |
| w/ Max-pooling | 2.18 | 1.37 | 48.67 | 7.84 | 5.33 | 15.05 | 63.63 | 48.99 | 8.61 |
| w/ Attention-weighted | 1.46 | 0.89 | 59.01 | 5.53 | 3.74 | 18.20 | 48.40 | 39.22 | 10.17 |
| w/ Mean-pooling (Ours) | 1.45 | 0.91 | 58.93 | 5.50 | 3.70 | 18.25 | 37.31 | 29.40 | 10.78 |

potential as a unified and systematic diagnostic tool. By sequentially tracking and thresholding the divergence of the hidden states of each error sample at different layers, we can pinpoint the specific point at which the internal representation starts to deviate significantly from the approximated correct reasoning trajectory.

## 4.5 ABLATION STUDY

We conducted an ablation study on three models in two tasks to compare the effectiveness of different aggregation strategies. We evaluated four methods: (1) Mean-pooling; (2) Last-token representation; (3) Max-pooling; and (4) Attention-weighted pooling. As shown in Table 2, Mean-pooling and the more complex Attention-weighted pooling consistently emerge as the top-performing methods, uniformly achieving the highest t-stat values. In contrast, the performance of Last-token is slightly inferior, likely because the representation of the last token is heavily optimized for the immediate task of predicting the next (often end-of-sequence) token, rather than serving as a global summary of the entire reasoning chain. While the final token's state is crucial for the ultimate decision, it may have already discarded important contextual information from earlier steps in the reasoning process, information that is equally vital for comprehensively characterizing the geometric structure of the reasoning state. The Max-pooling is the most unstable, as its lossy by retaining only the maximum activation value for each dimension, it may discard the subtle combination of features necessary for nuanced reasoning. This aggressive information filtering can disrupt the fine-grained geometric structure of the manifold, leading to an unstable or less clear separation between correct and erroneous reasoning states. Given that Mean-pooling, with its parameter-free and highly efficient nature, achieves top-tier performance nearly on par with its more complex counterpart, we conclude that it represents the most prudent and well-justified choice for our framework.

Crucially, across all combinations of models, tasks, and aggregation strategies tested, the deviation distance of error representations (Err Dist) remains significantly greater than the internal distance of correct representations (Corr Dist). All methods yield highly statistically significant t-stat values. This provides strong evidence that the core phenomenon that erroneous reasoning geometrically deviates from the correct reasoning manifold is robust and not an artifact of our specific choice of the mean-pooling method.

## 5 CONCLUSION

We introduced REMA, a novel interpretability framework for LLMs, centered on the concept of the "Reasoning Manifold". The experiments provide strong empirical evidence supporting the existence of such manifolds, and demonstrate that deviations from the manifold are highly indicative of reasoning errors, allowing for the localization of potential failure points (divergence points) within the model's inference path. Future directions include leveraging insights from manifold analysis to actively intervene and improve models, for instance, by designing novel regularization terms that encourage models to learn reasoning manifolds with more desirable geometric properties or by employing editing techniques to "pull back" deviating representations onto the manifold to correct errors. REMA presents a highly valuable and promising direction for future exploration.

## ETHICS STATEMENT

This research adheres to the ICLR Code of Ethics. The core objective of our work is to enhance the understanding of failure mechanisms in LLMs, thereby contributing to the scientific foundation for building more reliable and safer AI systems. All datasets and models used in our study are publicly available, sourced to promote open science and reproducible research. We have not used proprietary data or any data containing sensitive personal information.

## REPRODUCIBILITY STATEMENT

To ensure the reproducibility of our research, we have made the following efforts: All models used in our study, including their specific versions and weights, were sourced from public repositories (e.g., Hugging Face), with detailed information listed in Appendix C. All datasets are also public benchmarks; a complete description of their sources, the specific splits we used, and any preprocessing steps can be found in Appendix B. To further facilitate reproducibility, we will make all code used to generate the experimental results and figures in this paper publicly available via an anonymized link upon acceptance. The code repository will include the complete implementation for data preprocessing, hidden state extraction, the REMA analysis pipeline, and our sensitivity analyses.

## STATEMENT ON LLM USAGE

All core research ideas, experimental design, data analysis, interpretation of results, and the final scientific conclusions presented in this paper were conceived and formulated by the human authors. The LLM's role was strictly that of an assistant for language and expression, not a contributor to the core scientific content or intellectual contributions of the work. The authors take full responsibility for all content in this manuscript, including any text that was assisted by an LLM.

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

Table 3: Zero-shot success rates (%) of evaluated LLMs on reasoning tasks. Samples were subsequently partitioned into $\mathcal{D}_{\text{correct}}$ and $\mathcal{D}_{\text{error}}$ for analysis.

| | | Text Reasoning Tasks | | | Multimodal Reasoning Tasks | | | |
|---|---|---|---|---|---|---|---|---|
| | Model | GPQA (448) | GSM8K (1319) | MATH (500) | MathVista (1000) | SNLI-VE (2000) | VQAv2 (2000) | AI2D (3088) |
| **LLM** | Qwen3 (4B) | 26.4% | 22.3% | 17.4% | – | – | – | – |
| | Llama3.2 (3B) | 20.6% | 10.3 % | 11.6% | – | – | – | – |
| | Qwen3 (30B-A3B) | 32.9% | 35.0% | 28.0% | – | – | – | – |
| **MLLM** | Llama3.2 (11B) | 23.3% | 12.2% | 11.8% | 36.9% | 76.6% | 66.1% | 71.2% |
| | LLaVA-OneVision (7B) | 23.4% | 14.6% | 13.6% | 53.7% | 93.4% | 71.8% | 72.0% |
| | Qwen2.5-VL (3B) | 23.0% | 14.3% | 11.4% | 55.8% | 97.8% | 66.4% | 73.5% |
| | Llama3.2 (90B) | 31.8% | 24.3% | 14.6% | 47.2% | 96.3% | 72.2% | 79.4% |
| | Qwen2.5-VL (72B) | 32.7% | 32.2% | 16.2% | 70.8% | 97.2% | 70.6% | 86.1% |

Emmanuelle Salin, Badreddine Farah, Stéphane Ayache, and Benoit Favre. Are vision-language transformers learning multimodal representations? a probing perspective. In *Proceedings of the AAAI Conference on Artificial Intelligence*, volume 36, pp. 11248–11257, 2022.

Jiajun Song and Yiqiao Zhong. Uncovering hidden geometry in transformers via disentangling position and context. *arXiv preprint arXiv:2310.04861*, 2023.

Peiyang Song, Pengrui Han, and Noah Goodman. A survey on large language model reasoning failures. In *2nd AI for Math Workshop@ ICML 2025*, 2025.

Aaquib Syed, Can Rager, and Arthur Conmy. Attribution patching outperforms automated circuit discovery, 2023. URL https://arxiv.org/abs/2310.10348.

Zhiquan Tan, Chenghai Li, and Weiran Huang. The information of large language model geometry. *arXiv preprint arXiv:2402.03471*, 2024.

Ian Tenney, Dipanjan Das, and Ellie Pavlick. Bert rediscovers the classical nlp pipeline, 2019. URL https://arxiv.org/abs/1905.05950.

Ashish Vaswani, Noam Shazeer, Niki Parmar, Jakob Uszkoreit, Llion Jones, Aidan N. Gomez, Lukasz Kaiser, and Illia Polosukhin. Attention is all you need, 2023.

Ning Xie, Farley Lai, Derek Doran, and Asim Kadav. Visual entailment: A novel task for fine-grained image understanding, 2019. URL https://arxiv.org/abs/1901.06706.

An Yang, Anfeng Li, Baosong Yang, Beichen Zhang, Binyuan Hui, Bo Zheng, Bowen Yu, Chang Gao, Chengen Huang, Chenxu Lv, et al. Qwen3 technical report. *arXiv preprint arXiv:2505.09388*, 2025.

Zeping Yu and Sophia Ananiadou. Understanding multimodal llms: the mechanistic interpretability of llava in visual question answering, 2024. URL https://arxiv.org/abs/2411.10950.

Zeyu Yun, Yubei Chen, Bruno Olshausen, and Yann LeCun. Transformer visualization via dictionary learning: contextualized embedding as a linear superposition of transformer factors. In Eneko Agirre, Marianna Apidianaki, and Ivan Vulić (eds.), *Proceedings of Deep Learning Inside Out (DeeLIO): The 2nd Workshop on Knowledge Extraction and Integration for Deep Learning Architectures*, pp. 1–10, Online, June 2021. Association for Computational Linguistics. doi: 10. 18653/v1/2021.deelio-1.1. URL https://aclanthology.org/2021.deelio-1.1/.

Shaolei Zhang, Tian Yu, and Yang Feng. Truthx: Alleviating hallucinations by editing large language models in truthful space, 2024. URL https://arxiv.org/abs/2402.17811.

## A  BACKGROUND

To analyze the structure of the internal representation space of LLMs, we introduce the key metrics Intrinsic Dimension (ID) and Mutual Information (MI). They provide the theoretical foundations for our subsequent approximation, characterization, and analysis of the "Reasoning Manifold".

**Intrinsic Dimension (ID)** denoted as $d_{int}$, which describes the local effective degrees of freedom of a dataset embedded in a high-dimensional space (Fefferman et al., 2016). Unlike the ambient dimension $d$ of the space where the data reside, the intrinsic dimension reflects the minimum number of parameters needed to generate the data. In our research, we used ID to quantitatively validate the hypothesis that the representation space possesses a low-dimensional structure. We estimate the ID by analyzing the distribution of the ratio $\mu = r_2/r_1$, where $r_1$ and $r_2$ are the distances from each data point to its first and second nearest neighbors, respectively (Facco et al., 2017). For a point in a $d_{int}$-dimensional manifold, the cumulative distribution function (CDF) of $\mu$ follows $F(\mu) = 1 - \mu^{-d_{int}}$. The estimate of $d_{int}$ is then obtained by a linear regression fit to the empirical CDF $\hat{F}(\mu)$ according to $\log(1 - \hat{F}(\mu)) = -d_{int} \log(\mu)$. This method demonstrates good robustness for sparse, high-dimensional data. An estimated $d_{int}$ that is significantly smaller than $d$ provides initial geometric evidence for our Reasoning Manifold hypothesis.

**Mutual Information (MI)** is used to measure the statistical dependence between the representation $\mathbf{z}^l$ and the task objective $Y$, and is defined as

$$I(Z;Y) = H(Z) - H(Z|Y), \tag{3}$$

which represents the reduction in the uncertainty of $Z$ given the knowledge of $Y$. Since the true underlying distributions are generally unknown, directly computing MI is intractable. To this end, we adopt the $k$-nearest neighbor based Kraskov-Stögbauer-Grassberger (KSG) estimator (Kraskov et al., 2004) as a non-parametric approximation of MI. Given $N$ sample pairs $\{(\mathbf{z}_j, y_j)\}_{j=1}^{N}$, the first estimator (KSG-1) is given by:

$$\hat{I}(Z;Y) = \psi(k) - \langle \psi(n_z + 1) + \psi(n_y + 1) \rangle + \psi(N), \tag{4}$$

where $N$ is the number of samples, $k$ is the number of neighbors, and $\langle \cdot \rangle$ denotes the average over all samples. For each sample $i$, $\epsilon_i$ denotes the distance to its $k$-th nearest neighbor in the joint space $(Z, Y)$ under the maximum norm. The quantities $n_{z,i}$ and $n_{y,i}$ represent the number of neighbors strictly within a hypercube of side length $2\epsilon_i$ centered at sample $i$ in the marginal spaces $Z$ and $Y$, respectively. Through this neighborhood counting strategy, the KSG estimator provides a practical and distribution-free estimate of MI.

## B  DATASETS

To comprehensively assess the analytical capabilities of REMA across different reasoning scenarios, we carefully selected a total of seven benchmark datasets, covering both text reasoning and multimodal reasoning. The selection aims to encompass a diverse spectrum of reasoning abilities, from general visual question answering to complex multi-step mathematical deduction, specifically including text reasoning tasks:

**GSM8K** (Cobbe et al., 2021) [1], a dataset of 8.5K high quality linguistically diverse grade school math word problems. We find that even the largest transformer models fail to achieve high test performance. For our study, we used the test subset, which contains 1k test problems.

**MATH** (Hendrycks et al., 2021) [2], a dataset of 12,500 challenging competition mathematics problems. Each problem in MATH has a full step-by-step solution which can be used to teach models to generate answer derivations and explanations. Our datasets contain a subset of 500 problems from the MATH benchmark that OpenAI created.

**GPQA** (Rein et al., 2024) [3], a challenging dataset of 448 multiple-choice questions written by domain experts in biology, physics, and chemistry.

---

[1] https://github.com/openai/grade-school-math

[2] https://huggingface.co/datasets/HuggingFaceH4/MATH-500

[3] https://github.com/idavidrein/gpqa

Multimodal reasoning tasks:

**VQAv2** (Goyal et al., 2017) [4], a widely used benchmark for general-purpose visual question answering. It is a new dataset containing open-ended questions about images. These questions require an understanding of vision, language and commonsense knowledge to answer. For our experiments, we utilized a subset of 2,000 questions from the official validation split, ensuring each question corresponded to a unique image to maintain diversity.

**SNLI-VE** (Xie et al., 2019) [5], designed for the visual entailment task, requiring models to determine the relationship (entailment or contradiction) between an image and a given textual hypothesis. It is built on top of SNLI and Flickr30K. The problem that VE is trying to solve is to reason about the relationship between an image premise and a text hypothesis. Following official recommendations for a binary VE task, we excluded samples labeled as "neutral" retaining only those with entailment or contradiction labels. Our final evaluation set for SNLI-VE comprised 2,000 instances.

**MathVista** (Lu et al., 2024) [6], a challenging benchmark for evaluating models' visual mathematical reasoning abilities. It consists of three newly created datasets, IQTest, FunctionQA, and PaperQA, which address the missing visual domains and are tailored to evaluate logical reasoning on puzzle test figures, algebraic reasoning over functional plots, and scientific reasoning with academic paper figures, respectively. For our study, we used the test mini split, which contains 1,000 carefully curated question-image pairs.

**AI2D** (Kembhavi et al., 2016) [7], a question answering dataset focused on understanding diagrams and charts. It is comprised of about 5000 diagrams representing topics from grade school science, each annotated with constituent segmentations and their relationships. We utilized a subset of approximately 3.09K question-diagram pairs for our study.

## C  MODELS

To validate the effectiveness of REMA across different model architectures and parameter sizes, we selected a range of widely recognized and used by academia and industry, open-source LLMs and MLLMs. Our selection covers multiple model families and parameter sizes ranging from 3B to 90B.

These models are:

**Qwen3 (4B, 30B)** (Yang et al., 2025) [8] is the latest version of the Qwen model family. Qwen3 comprises a series of LLMs designed to advance performance, efficiency, and multilingual capabilities.

**Qwen2.5-VL (3B, 72B)** (Bai et al., 2025) [9] is the new flagship vision-language model of Qwen. The key features include: Understand things visually, Capable of visual localization in different formats, Generating structured outputs. It has significantly enhanced its general image recognition capabilities, expanding the categories of images to an ultra-large number.

**Llama3.2(3B, 11B, 90B)** (Grattafiori et al., 2024) [10] is built on top of Llama 3.1. The Llama3.2-Vision instruction-tuned models are optimized for visual recognition, image reasoning, captioning, and answering general questions about an image. The models outperform many of the available open source and closed multimodal models on common industry benchmarks.

**LLaVA-OneVision (7B)** (Li et al., 2024) [11] is an open-source multimodal LLM trained by fine-tuning Qwen2 on GPT-generated multimodal instruction-following data. LLaVA-OneVision-7B Chat was added in September 2024.

To establish a clear performance baseline and an overview of our data for subsequent interpretability analyses, we first evaluated the zero-shot accuracy of the selected LLMs on text and multimodal

---

[4]https://visualqa.org/index.html

[5]https://github.com/necla-ml/SNLI-VE

[6]https://huggingface.co/datasets/AI4Math/MathVista

[7]https://huggingface.co/datasets/lmms-lab/ai2d

[8]https://huggingface.co/Qwen/Qwen3-4B-Instruct-2507

[9]https://huggingface.co/Qwen/Qwen2.5-VL-3B-Instruct

[10]https://huggingface.co/meta-llama/Llama-3.2-11B-Vision-Instruct

[11]https://huggingface.co/llava-hf/llava-onevision-qwen2-7b-ov-chat-hf

Table 4: Average deviation distance on larger and Mixture-of-Experts (MoE) models. The results confirm that the geometric deviation of error states remains a robust phenomenon in larger models.

| Model | Task | Avg Error Dist. | Avg Correct Dist. | T-stat |
|---|---|---|---|---|
| Llama3.2 (90B) | SNLI-VE | 2.93 | 1.79 | 29.96 |
| Qwen2.5-VL (72B) | SNLI-VE | 25.36 | 14.18 | 22.10 |
| Qwen3-30B-A3B (MoE) | GPQA | 7.78 | 6.46 | 8.64 |
| Llama3.2 (11B) | SNLI-VE | 1.45 | 0.91 | 58.93 |
| Qwen2.5-VL (3B) | SNLI-VE | 5.50 | 3.70 | 18.25 |
| Qwen3 (4B) | GPQA | 38.40 | 30.22 | 8.87 |

reasoning benchmark datasets. Based on this evaluation, the samples were partitioned for further study. Table 3 presents the accuracy resulting from each model in the all benchmark datasets. All models performed zero-shot inference.

For all experiments, we used the official pre-trained weights and employed their default zero-shot inference configurations (e.g., $temperature$, $top\_k$, $top\_p$).

## D  IMPLEMENTATION DETAILS

All experiments were carried out on NVIDIA A100 (40G) GPUs, using CentOS Linux 7.5 and Python 3.9.12. The core deep learning framework was PyTorch 2.1.0, with the Transformers library version 4.50.0.

In our REMA framework, we primarily analyze the mean-pooled hidden states (as defined in Equation 1) from each decoder layer within the model's language module. For the estimation of intrinsic dimensions, we employed the TwoNN estimator (Facco et al., 2017). For task-relevant mutual information computation with text answers, we converted ground truth answers into fixed-dimensional embedding vectors using a Sentence-BERT model (all-MiniLM-L6-v2) (Reimers & Gurevych, 2019). We then used the $k$-nearest neighbor-based KSG mutual information estimator (Kraskov et al., 2004) (implemented via the NPEET toolkit), also with $k = 5$. When calculating the deviation distances (Section 3.3), we used the Euclidean distance and the nearest neighbors $k' = 5$. For the separability test (Section 3.3), we used an SVC with an RBF kernel, evaluating accuracy via 5-fold cross-validation. In localizing divergence points (Section 3.4), the threshold factor $\alpha$ was set to 2.0, and the number of neighbors $k'$ for $D_j^l$ computation was 5. For UMAP visualizations, we configured $n\_neighbors = 5$, $min\_dist = 0.1$, $spread = 1.0$, and used the cosine distance as the metric.

## E  LIMITATIONS AND DISCUSSION

While our REMA framework offers novel perspectives and valuable insights into LLM reasoning failures, the current study has several limitations, which in turn open up multiple avenues for future research. First, we use the point cloud of correct samples as an empirical approximation of the manifold, the fidelity of which depends on the diversity and density of these samples. Future work could build more precise manifold models using advanced manifold learning techniques and further investigate the fundamental reasons for the formation of these structures, such as their relationship to task structure or model inductive biases. Second, Our partitioning of reasoning samples into "correct" and "error" sets is based on exact match of the answer text. This binary partitioning might misclassify "nearly correct" reasoning as erroneous, or vice versa. Future work could explore more fine-grained, confidence-based, or soft-scoring-based partitioning methods.

# F  EXPERIMENTAL RESULTS

## F.1  SCALING ANALYSIS ON LARGER LLMS

To validate the generalizability of our findings to larger LLms, we conducted additional validation experiments on three models with significantly larger parameter or advanced architectures. Specifically, we analyzed Llama3.2 (90B) and Qwen2.5-VL (72B) on the SNLI-VE, , comparing them with their smaller counterparts, **Llama3.2 (11B)** and **Qwen2.5-VL (3B)**, respectively. Additionally, we analyzed a Mixture-of-Experts (MoE) model, **Qwen3 (30B-A3B)**, on the GPQA , referencing it against a similarly-sized dense model, **Qwen3 (4B)**. We applied the same methodology as in the main experiments, computing the deviation distances averaged across all decoder layers.

The results, presented in Table 4, demonstrate that our core findings remain robust and generalizable. For all three larger models, the average deviation distance of error samples is consistently and significantly greater than the average internal distance of correct samples, as indicated by the high t-statistics (ranging from 8.64 to 29.96). For instance, the Llama3.2 model exhibits a clear deviation phenomenon ($D_{\text{error}} = 2.93$ vs. $D_{\text{correct}} = 1.79$), as does the Qwen2.5-VL model ($D_{\text{error}} = 25.36$ vs. $D_{\text{correct}} = 14.18$).

More interestingly, by comparing models of different scales, we can observe some initial indications of scaling effects: For the Llama3.2-Vision series (on SNLI-VE): Scaling from 11B to 90B parameters, we observe that the absolute values of both $D_{\text{error}}$ (from 1.45 to 2.93) and $D_{\text{correct}}$ (from 0.91 to 1.79) increase. However, their relative deviation (e.g., the ratio $D_{\text{error}}/D_{\text{correct}}$) remains at a similar level (approx. 1.59 for 11B and 1.64 for 90B). This may indicate that as model scale increases, its representation space becomes larger or sparser, leading to a corresponding increase in all distances, while the relative geometric relationship between the correct manifold and the error regions is preserved. For the Qwen2.5-VL series (on SNLI-VE): Scaling from 3B to 72B, we see a similar, even more pronounced, increase in the absolute values of both $D_{\text{error}}$ (from 5.50 to 25.36) and $D_{\text{correct}}$ (from 3.70 to 14.18), again suggesting that larger models may operate in representation spaces of a larger scale. The relative deviation also remains in a comparable range ($D_{\text{error}}/D_{\text{correct}}$ is approx. 1.49 for 3B and 1.79 for 72B). For the Qwen family (on GPQA): Although the Qwen3 (30B-A3B) is an MoE model and its parameter is not directly comparable to dense models, we can reference it against the Qwen3 (4B) model. We find that the larger, more advanced MoE model exhibits far smaller absolute deviation distances ($D_{\text{error}} = 7.78, D_{\text{correct}} = 6.46$) than the smaller dense model ($D_{\text{error}} = 38.40, D_{\text{correct}} = 30.22$). This strongly suggests that the MoE architecture may learn a geometrically more compact representation space, even while potentially achieving better performance (as seen in Table 3).

These validation experiments provide strong evidence that the core phenomenon, that erroneous reasoning geometrically deviates from the manifold of correct reasoning, is not confined to the models analyzed in the main text, but exhibits generalizability across a wider range of model and architectural types, significantly strengthening the credibility and applicability of the REMA framework.

## F.2  t-SNE VISUALIZATION OF DEVIATIONS

To complement the UMAP visualizations presented in Figure 2, we additionally performed t-SNE projection of the final-layer hidden states of all three MLLMs on the SNLI-VE task. The resulting plots are shown in Figure 5.

Consistent with the UMAP results, the t-SNE plots reveal a clear spatial separation between the representations of correct and error reasoning samples. First, correct samples (blue) tend to form tight and coherent clusters, while error samples (red) are distributed more diffusely and often reside on the periphery or in distinct, less structured clusters. Second, across all three models (Llama3.2, LLaVA-OneVision, and Qwen2.5-VL), this separation is qualitatively stable, confirming the robustness of the deviation patterns across different nonlinear projection methods. This agreement between t-SNE and UMAP projections supports our claim that error representations systematically diverge from the manifold formed by correct reasoning, and such divergence is geometrically significant in the latent space.

Furthermore, to investigate how this separation develops layer-wise, we conducted a layer-by-layer t-SNE visualization for Llama3.2 on the SNLI-VE task. The progression is shown in Figure 6,

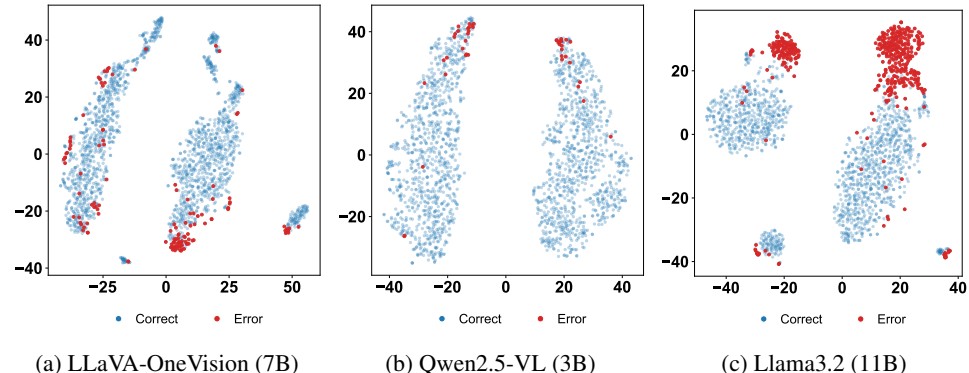

Figure 5: t-SNE visualization of hidden states from the final decoder layer on the SNLI-VE for three LLMs. Consistent with the UMAP results, the t-SNE plots reveal a clear spatial separation between the representations of correct and error reasoning samples.

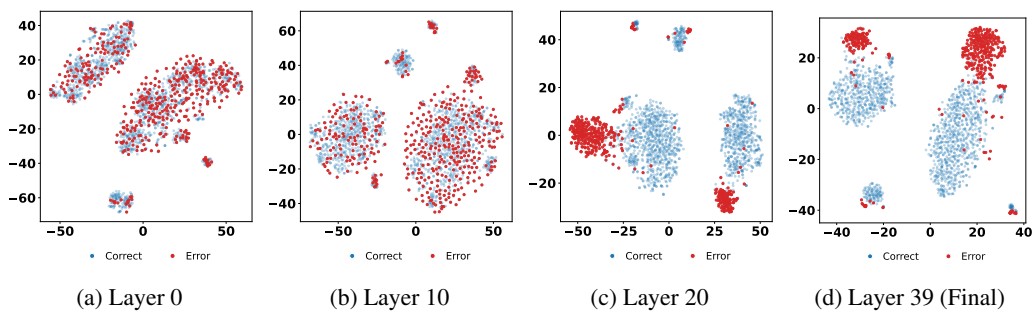

Figure 6: t-SNE visualization for Llama3.2 (11B) from the first layer (Layer 0) to final layer (Layer 39) on SNLI-VE. In the early layers, representations of correct and error samples are heavily intermingled, indicating that the model has not yet differentiated between successful and failed reasoning trajectories. As depth increases, the clusters become more structured and around the mid-to-late layers, the representations of error samples begin to deviate and spread away from the core regions occupied by correct samples. By the final layers, the separation becomes prominent.

covering layers from the first decoder layer (Layer 0) to the final decoder layer (Layer 39). We observed a clear evolutionary pattern:

**(a) Early Layers (Layer 0):** In the model's early layers, representations of correct (blue) and error (red) samples are heavily intermingled. This indicates that the model has not yet effectively differentiated between successful and failed reasoning trajectories in its representation space.

**(b) Mid Layers (Layer 10, Layer 20):** As layer depth increases, cluster structures begin to emerge and become progressively clearer. Particularly in the mid-layers (as seen in Figures 6b and 6c), the representations of error samples start to gradually "peel away" from the core regions occupied by correct samples, moving towards the cluster peripheries or forming independent, less-structured smaller clusters.

**(c) Last Layers (Layer 39):** By the final layers of the model (Figure 6d), the separation becomes highly pronounced. Correct samples form tight and well-defined clusters, while error samples are clearly pushed to the outskirts of these clusters or into entirely separate regions. This final-layer pattern is highly consistent with the UMAP visualizations presented in the main paper (Figure 2).

This layer-wise progression provides a direct visual demonstration that the geometric divergence of erroneous reasoning emerges and is amplified progressively during inference, becoming solidified in the deeper representations of the model. This observation is also highly consistent with the trend of increasing separability accuracy with layer depth, as shown in our quantitative analysis in the main paper (Figure 3). The consistency between t-SNE and UMAP results strengthens the validity of the

Table 5: Sensitivity analysis of average deviation distances to the number of neighbors $k'$ for Qwen3 (4B) on GPQA. Distances are averaged across samples for the specified layer. All differences between "Avg. Correct Dist." and "Avg. Error Dist." for a given $k'$.

| Layer | Avg. Correct Dist. ($D_{\text{correct}}^l$) | | | | Avg. Error Dist. ($D_{\text{error}}^l$) | | | |
|---|---|---|---|---|---|---|---|---|
| | $k' = 5$ | $k' = 10$ | $k' = 15$ | $k' = 20$ | $k' = 5$ | $k' = 10$ | $k' = 15$ | $k' = 20$ |
| 0 | 1.85 | 1.96 | 2.08 | 2.17 | 1.50 | 1.82 | 2.03 | 2.17 |
| 1 | 2.61 | 2.76 | 2.93 | 3.04 | 2.08 | 2.52 | 2.81 | 2.98 |
| 2 | 3.25 | 3.44 | 3.65 | 3.79 | 2.60 | 3.14 | 3.50 | 3.73 |
| 3 | 4.44 | 4.72 | 5.02 | 5.24 | 3.49 | 4.22 | 4.71 | 5.03 |
| 4 | 6.16 | 6.58 | 7.01 | 7.33 | 4.80 | 5.82 | 6.49 | 6.93 |
| 5 | 7.82 | 8.35 | 8.90 | 9.30 | 6.10 | 7.40 | 8.29 | 8.86 |
| 6 | 9.32 | 9.91 | 10.53 | 10.96 | 7.27 | 8.80 | 9.84 | 10.49 |
| 7 | 10.56 | 11.23 | 11.94 | 12.43 | 8.23 | 9.97 | 11.16 | 11.90 |
| 8 | 11.84 | 12.60 | 13.40 | 13.95 | 9.34 | 11.31 | 12.64 | 13.46 |
| 9 | 12.50 | 13.31 | 14.13 | 14.70 | 9.89 | 11.96 | 13.32 | 14.17 |
| 10 | 12.47 | 13.28 | 14.11 | 14.69 | 9.86 | 11.91 | 13.28 | 14.14 |
| 11 | 12.85 | 13.69 | 14.55 | 15.15 | 10.15 | 12.27 | 13.66 | 14.54 |
| 12 | 13.00 | 13.85 | 14.72 | 15.33 | 10.28 | 12.43 | 13.84 | 14.74 |
| 13 | 13.55 | 14.43 | 15.32 | 15.96 | 10.73 | 12.98 | 14.43 | 15.37 |
| 14 | 13.27 | 14.13 | 14.99 | 15.61 | 10.46 | 12.64 | 14.07 | 14.98 |
| 15 | 13.34 | 14.19 | 15.07 | 15.69 | 10.49 | 12.67 | 14.12 | 15.05 |
| 16 | 13.57 | 14.41 | 15.27 | 15.88 | 10.64 | 12.85 | 14.27 | 15.19 |
| 17 | 14.67 | 15.56 | 16.46 | 17.10 | 11.52 | 13.87 | 15.37 | 16.34 |
| 18 | 15.93 | 16.90 | 17.87 | 18.55 | 12.53 | 15.09 | 16.70 | 17.74 |
| 19 | 18.79 | 19.86 | 20.92 | 21.69 | 14.67 | 17.62 | 19.44 | 20.62 |
| 20 | 21.16 | 22.32 | 23.45 | 24.29 | 16.44 | 19.73 | 21.72 | 23.01 |
| 21 | 23.26 | 24.51 | 25.71 | 26.60 | 18.07 | 21.68 | 23.81 | 25.21 |
| 22 | 26.32 | 27.70 | 29.03 | 30.01 | 20.34 | 24.42 | 26.80 | 28.36 |
| 23 | 31.68 | 33.37 | 34.89 | 36.06 | 24.64 | 29.59 | 32.34 | 34.18 |
| 24 | 38.87 | 40.95 | 42.76 | 44.17 | 30.55 | 36.68 | 39.92 | 42.12 |
| 25 | 41.44 | 43.71 | 45.70 | 47.29 | 32.57 | 39.13 | 42.65 | 45.05 |
| 26 | 46.82 | 49.44 | 51.67 | 53.46 | 36.88 | 44.32 | 48.24 | 50.94 |
| 27 | 52.79 | 55.72 | 58.16 | 60.12 | 41.58 | 49.94 | 54.22 | 57.20 |
| 28 | 59.50 | 62.74 | 65.41 | 67.58 | 46.87 | 56.27 | 61.04 | 64.32 |
| 29 | 68.27 | 71.86 | 74.69 | 77.02 | 53.71 | 64.39 | 69.65 | 73.24 |
| 30 | 80.10 | 84.22 | 87.46 | 90.08 | 63.12 | 75.58 | 81.67 | 85.80 |
| 31 | 88.16 | 92.73 | 96.38 | 99.30 | 69.79 | 83.55 | 90.32 | 94.89 |
| 32 | 103.23 | 108.59 | 112.90 | 116.38 | 81.82 | 98.07 | 105.95 | 111.25 |
| 33 | 121.61 | 127.67 | 132.47 | 136.37 | 96.35 | 115.28 | 124.36 | 130.48 |
| 34 | 151.16 | 158.82 | 165.00 | 169.97 | 119.49 | 142.75 | 154.10 | 161.73 |
| 35 | 176.86 | 185.65 | 192.52 | 197.82 | 139.63 | 166.62 | 179.66 | 188.11 |

observed representational patterns and underscores the utility of dimensionality reduction techniques in diagnosing the reasoning behaviors of LLMs.

## F.3 SENSITIVITY AND ROBUSTNESS ANALYSIS OF REMA

### F.3.1 SENSITIVITY ANALYSIS OF DEVIATION DISTANCE TO NUMBER OF NEIGHBORS

To assess the robustness of our findings regarding the deviation of error representations from the correct reasoning manifold, we performed a sensitivity analysis on the number of nearest neighbors ($k'$) used in the deviation distance calculation. We recomputed the average internal distance for correct samples ($D_{\text{correct}}^l$) and the average deviation distance for error samples ($D_{\text{error}}^l$) using $k' \in \{5, 10, 10, 20\}$ for a representative LLM and task combination (Qwen3 (4B) on GPQA). Table 5 presents the results . The 'Avg. Correct Dist.' columns show $D_{\text{correct}}^l$ and the 'Avg. Error Dist.' columns show $D_{\text{error}}^l$ for each $k'$. Across all tested $k'$ values and layers, $D_{\text{error}}^l$ consistently remains

Table 6: Aggregated divergence point ($l_{\text{diverge}}$) counts for Qwen2.5-VL (3B), comparing threshold factors $\alpha = 1.0$ and $\alpha = 2.0$. Counts are summed over 8-layer intervals.

| Layer Interval | AI2D | | MathVista | |
|---|---|---|---|---|
| | $\alpha = 1.0$ | $\alpha = 2.0$ | $\alpha = 1.0$ | $\alpha = 2.0$ |
| 0-7 | 270 | 55 | 100 | 35 |
| 8-15 | 9 | 13 | 39 | 7 |
| 16-23 | 17 | 12 | 52 | 8 |
| 24-31 | 23 | 25 | 51 | 60 |
| 32+ | 12 | 10 | 2 | 1 |

significantly greater than $D_{\text{correct}}^l$. While the absolute values of the distances increase with larger $k'$, the crucial observation is that the systematic and statistically significant gap between the deviation of error states and the internal dispersion of correct states persists robustly. For instance, at Layer 20, the difference $D_{\text{error}}^l - D_{\text{correct}}^l$ remains substantial across all $k'$ values. This analysis indicates that our central finding mobilité the geometric deviation of erroneous reasoning states from the manifold of correct reasoning is not highly sensitive to the specific choice of $k'$ within a reasonable range, thereby strengthening the reliability of our conclusions. The choice of $k' = 5$ for the main experiments was selected as it provides a good balance between capturing local structure and maintaining stability against individual outliers.

### F.3.2 SENSITIVITY OF DIVERGENCE POINT LOCALIZATION TO THRESHOLD FACTOR

To assess the robustness of our divergence point localization (Section 3.4) to the choice of the threshold factor $\alpha$ (defined in Equation 2), we compared the distribution of divergence points $l_{\text{diverge}}$ using a lenient threshold of $\alpha = 1.0$ versus a stricter threshold of $\alpha = 2.0$ (the value used in our main experiments). This analysis was performed for the Qwen2.5-VL (3B) model on the AI2D and MathVista datasets. For a summarized view, the raw layer-wise divergence counts were aggregated into 8-layer intervals.

Table 6 presents the aggregated divergence point counts for these settings. As anticipated, employing a stricter threshold of $\alpha = 2.0$ generally reduces the number of identified divergence points, as it requires a more substantial deviation from the mean internal distance of correct samples to be classified as a divergence.

However, despite the variation in absolute counts, the analysis reveals robust and interesting patterns. For AI2D, the very early layers (Interval 0-7) remain the most frequent stage for divergence origination under both $\alpha$ settings, although the count decreases significantly with the stricter threshold. This suggests that for this task, a large number of failures begin with a relatively small initial deviation, while a substantial subset of failures still originate from more pronounced early-stage deviations.

More strikingly, for MathVista, increasing the threshold factor $\alpha$ from 1.0 to 2.0 shifts the primary peak of divergence points from the early layers (Interval 0-7, count drops from 100 to 35) to the mid-to-late layers (Interval 24-31, count increases from 51 to 60). This indicates that while many reasoning paths in MathVista start to stray with minor deviations in early layers, the most significant and severe deviations tend to manifest and be identified during the later, potentially more complex, stages of the reasoning process.

This sensitivity analysis demonstrates that while the precise number of identified divergence points is dependent on $\alpha$, the broader identification of early and mid-to-late processing stages as being susceptible to initial reasoning failures is robust. Furthermore, it reveals that adjusting $\alpha$ can help differentiate between failures originating from subtle, early deviations versus those originating from more severe, later-stage deviations, providing an even more granular diagnostic capability. Our choice of $\alpha = 2.0$ in the main experiments reflects a focus on identifying these more pronounced deviations.

### F.3.3 ROBUSTNESS OF DEVIATION DISTANCE TO SAMPLE SIZE PERTURBATIONS

To investigate the robustness of our deviation distance findings (Section 3.3) to variations in the number of samples used for analysis, we conducted a subsampling experiment. For the Qwen2.5-

Table 7: Sensitivity analysis of average deviation distances to sample size subsampling ratios for Qwen3 on GPQA. Distances are averaged across samples for each layer using 50%, 70%, or 100% of the available correct and error samples respectively.

| Layer | Avg. Correct Dist. ($D_{correct}^l$) | | | Avg. Error Dist. ($D_{error}^l$) | | |
|---|---|---|---|---|---|---|
| | @50% | @70% | @100% | @50% | @70% | @100% |
| 0 | 1.97 | 1.89 | 1.85 | 1.62 | 1.57 | 1.50 |
| 1 | 2.74 | 2.65 | 2.61 | 2.20 | 2.17 | 2.08 |
| 2 | 3.43 | 3.31 | 3.25 | 2.73 | 2.70 | 2.60 |
| 3 | 4.71 | 4.55 | 4.44 | 3.67 | 3.64 | 3.49 |
| 4 | 6.66 | 6.35 | 6.16 | 5.12 | 5.04 | 4.80 |
| 5 | 8.46 | 8.05 | 7.82 | 6.54 | 6.38 | 6.10 |
| 6 | 10.03 | 9.58 | 9.32 | 7.81 | 7.62 | 7.27 |
| 7 | 11.33 | 10.84 | 10.56 | 8.89 | 8.62 | 8.23 |
| 8 | 12.81 | 12.19 | 11.84 | 10.17 | 9.78 | 9.34 |
| 9 | 13.56 | 12.87 | 12.50 | 10.71 | 10.33 | 9.89 |
| 10 | 13.50 | 12.82 | 12.47 | 10.62 | 10.29 | 9.86 |
| 11 | 13.96 | 13.23 | 12.85 | 10.90 | 10.57 | 10.15 |
| 12 | 14.13 | 13.40 | 13.00 | 11.05 | 10.71 | 10.28 |
| 13 | 14.75 | 13.98 | 13.55 | 11.55 | 11.17 | 10.73 |
| 14 | 14.42 | 13.68 | 13.27 | 11.25 | 10.89 | 10.46 |
| 15 | 14.47 | 13.74 | 13.34 | 11.30 | 10.91 | 10.49 |
| 16 | 14.67 | 13.96 | 13.57 | 11.42 | 11.06 | 10.64 |
| 17 | 15.84 | 15.07 | 14.67 | 12.36 | 11.96 | 11.52 |
| 18 | 17.18 | 16.37 | 15.93 | 13.44 | 13.00 | 12.53 |
| 19 | 20.10 | 19.27 | 18.79 | 15.72 | 15.21 | 14.67 |
| 20 | 22.56 | 21.66 | 21.16 | 17.58 | 17.01 | 16.44 |
| 21 | 24.82 | 23.85 | 23.26 | 19.37 | 18.71 | 18.07 |
| 22 | 28.04 | 27.01 | 26.32 | 21.80 | 21.08 | 20.34 |
| 23 | 33.88 | 32.65 | 31.68 | 26.37 | 25.58 | 24.64 |
| 24 | 41.64 | 40.11 | 38.87 | 32.88 | 31.81 | 30.55 |
| 25 | 44.39 | 42.73 | 41.44 | 34.94 | 33.80 | 32.57 |
| 26 | 50.35 | 48.36 | 46.82 | 39.60 | 38.26 | 36.88 |
| 27 | 56.75 | 54.59 | 52.79 | 44.50 | 43.10 | 41.58 |
| 28 | 63.68 | 61.41 | 59.50 | 49.84 | 48.41 | 46.87 |
| 29 | 73.16 | 70.39 | 68.27 | 56.85 | 55.28 | 53.71 |
| 30 | 85.52 | 82.38 | 80.10 | 66.39 | 64.68 | 63.12 |
| 31 | 94.22 | 90.67 | 88.16 | 73.24 | 71.31 | 69.79 |
| 32 | 110.61 | 106.25 | 103.23 | 85.98 | 83.60 | 81.82 |
| 33 | 130.05 | 125.09 | 121.61 | 100.75 | 98.25 | 96.35 |
| 34 | 161.50 | 155.54 | 151.16 | 124.45 | 121.66 | 119.49 |
| 35 | 188.94 | 182.03 | 176.86 | 144.09 | 141.66 | 139.63 |

VL(3B) on GPQA, we randomly subsampled 50%, 70%, and 100% (all available samples) of both the correct ($\mathcal{Z}_{correct}^l$) and error ($\mathcal{Z}_{error}^l$) representation sets at each layer. We then recomputed the average internal distance for correct samples ($D_{correct}^l$) and the average deviation distance for error samples ($D_{error}^l$) using these subsets.

Table 7 presents the layer-wise average distances for each subsampling ratio. The results reveal a highly stable and intriguing pattern: across all layers and all tested subsampling ratios, the average internal distance of correct samples ($D_{correct}^l$) is consistently greater than the average internal distance of error samples ($D_{error}^l$). For example, at Layer 35, $D_{correct}^l$ is approximately 176.86 while $D_{error}^l$ is 139.63 (at 100% data), and this relationship holds even with 50% of the data.

This observation, while contrary to the simple intuition that error states should be more dispersed, provides a deeper insight into a potential failure mode for this model-task combination. It suggests that for GPQA, a complex knowledge-intensive task, the correct reasoning process may require

exploring a more diverse and expansive representational space (the "reasoning manifold") to integrate various pieces of information, resulting in a larger internal distance ($D_{\text{correct}}^l$). Conversely, erroneous reasoning might correspond to the model "collapsing" its representation into a more compact, lower-variance region, possibly by prematurely converging on an incorrect but high-confidence answer based on simpler heuristics. This aligns with our findings on other metrics (e.g., intrinsic dimension or entropy) where correct states sometimes exhibit higher complexity than error states.

Crucially, this experiment still demonstrates the robustness of our overall framework. The fact that there is a consistent and significant difference between the geometric properties (in this case, internal dispersion) of correct and erroneous representations is the core finding. It validates that these two sets of states occupy structurally distinct regions in the representation space, even if the nature of that distinction (e.g., which set is more dispersed) is task- and model-dependent. The stability of this finding across different sample sizes confirms that it is not an artifact of the full dataset but a robust characteristic of the model's reasoning process on this task.

