# OpenReview forum: "REMA: A Unified Reasoning Manifold Framework for Interpreting Large Language Model"
_ICLR.cc/2026/Conference — ICLR 2026 Conference Withdrawn Submission_

### Official Review · Reviewer_g48d · 2025-10-26

**Soundness:** 3
**Presentation:** 3
**Contribution:** 3
**Rating:** 6
**Confidence:** 3

**Summary:**

This paper introduces “Reasoning Manifold” as a low-dimensional latent structure in the internal representations of LLMs corresponding to all correct reasoning paths.
Building on this, the authors propose REMA, a post-hoc framework to identify reasoning failures by quantifying geometric deviations of erroneous hidden states from this manifold (via k-NN distances), and localizing divergence points across layers (via statistical thresholding).
Experiments on text and multimodal tasks with various models show: (1) the low-dimensional nature of the reasoning manifold, (2) significant separability between correct and error representations, and (3) task/model-dependent divergence patterns.

**Strengths:**

1. The paper is clearly structured and innovative. The authors provide a new geometric perspective on interpretability by introducing reasoning manifolds.
2. The experiments are comprehensive, covering 7 benchmarks and 8 models. The visualization of the experiments also enhances the credibility of the paper.

**Weaknesses:**

The experimental details in this paper need further explanation, please see the Questions section.

Minor comments:
- Line 157: next token $y_{i, t}$ -> $y_{i, t+1}$
- Line 426 claims that "we aggregated consecutive 8 decoder layers into a single bin" but this does not seem to be the case in Figure 4. Furthermore, (a) and (c) appear to be the same model (Llama3.2 11B), and (b) and (d) appear to be the same model (Qwen3 4B).

**Questions:**

1. MI is defined as the reduction in the uncertainty of Z given the knowledge of Y, and the line 317-319 claims “the representations of correct reasoning paths contain more useful information pertinent to the correct answer from the initial stages” but there is no explanation as to why the amount of uncertainty reduction is related to the amount of useful information about the correct answer at the initial stages.
2. In Figure 2, the authors present a UMAP visualization for the SNLI-VE task. Table 1 shows that this benchmark is a simple task, with all three models achieving over 76% accuracy. This means that the number of incorrect samples in Figure 2 is much smaller than the number of correct samples, so I'm curious whether the visualizations for more difficult tasks can still provide good discrimination.
3. Lines 372-373 claim "Correct samples often form one or more relatively concentrated clusters." Did the authors conduct further analysis to analyze the connections between different clusters?

---

### Official Review · Reviewer_7X3W · 2025-10-28

**Soundness:** 1
**Presentation:** 3
**Contribution:** 2
**Rating:** 4
**Confidence:** 4

**Summary:**

This paper proposes REMA, a interpretability framework that aims to understand the reasoning process of LLMs and multimodal LLMs from a manifold perspective.

**Strengths:**

1. The overall idea of modeling the reasoning from the perspective of hidden state's manifold sounds interesting.
2. The authors have taken a significant empirical effort, employing tools such as Intrinsic Dimension (ID) estimation (using TwoNN) and Mutual Information (MI) estimation (using KSG) to characterize the properties of these latent spaces. The authors also conducted varieties of ablation studies like the impact of pooling.
3. Although code is not provided, details in the appendix supports reproducibility,

**Weaknesses:**

Here are two weaknesses I believe that are substantial for the paper:

**Fundamental Mismatch Between "Theoretical" Claim and Actual Methodology**: The most significant weakness of this paper I think is the clear disconnect between its central claim of a "theoretical" investigation into reasoning manifolds and its overwhelmingly empirical approach. The title and abstract often allude to theoretical insights, yet the paper hasn't provided even one theorem. Concepts central to manifold theory, such as connectivity, curvature, geodesics, tangent spaces, or even a formal definition of what constitutes a "reasoning manifold" in the context of LLM latent spaces, **are entirely absent**. Honestly speaking, I was thrilled to when I saw the title and abstract at first but was a bit disappointed by the actual content of the paper. Using terms like "manifold" without any rigorous mathematical foundations or connection to established manifold theory is clearly misleading.

I understand it may be difficult to establish a complete manifold theory, but I suggest the authors should either modify their central claim or incorporate more theoretical foundations before publication.

**Flawed Answer Correctness Evaluation**: Using strict string matching to determine answer correctness does not make sense, especially for datasets like MATH. For example, `(x+1)(x-1)` and `x^2-1` are semantically identical answers in algebra, but strict string matching would label them as different. The authors can apply some evaluation resources from the community like EleutherAI's lm-evaluation-harness.

**Questions:**

Besides the weaknesses above, I also have some additional concerns and questions.

1. Do you believe 2NN-based ID and KSG MI estimators are accurate enough for the estimation? They may be unstable when the data is sparse or in high-dimensional spaces.
2. Could you provide justifications on the choice of some hyperparameters, like $\alpha = 2$ for layer-wise divergence localization?
3. What contributed to the formation of the low-dimensional "manifold"? Is model architecture or training methodology connected to this phenomenon?
4. Are the properties of the "reasoning manifold" consistent across models and datasets? Table 1 shows that actually the ID may be very different across different datasets.
5. The authors only performed binary classification on correctness, which is mentioned in the limitations section. Did you try to classify the answers like according to the failure modes?
6. Did you test your methods on other practical domains like coding?
7. Could your findings be applied to modify LLMs' behavior or enhance their reliability or reasoning abilities? This is discussed in the future work section but I think some small-scale preliminary experiments would be desirable.
8. There is a formatting problem that Table 3 appears to be within the references part.

---

### Official Review · Reviewer_YLdF · 2025-10-28

**Soundness:** 2
**Presentation:** 2
**Contribution:** 2
**Rating:** 4
**Confidence:** 3

**Summary:**

The paper proposes REMA, a unified framework that interprets LLM reasoning through the concept of a Reasoning Manifold: a low-dimensional geometric structure representing correct reasoning paths.
By quantifying geometric deviations between correct and erroneous representations, REMA identifies where reasoning begins to fail within the model.
This approach offers a measurable and interpretable way to analyze reasoning failures, supported by extensive experiments across various LLMs and multimodal models.

**Strengths:**

1. This paper provides a clear and direct method to localize reasoning failures within model layers through measurable spatial deviations.
2. Extensive experiments across different models and tasks validate the framework’s robustness and general applicability.

**Weaknesses:**

1. The authors’ partitioning of the reasoning manifold seems somewhat **coarse**. They simply divide each task’s dataset into a correct set $D_{correct}$ and an error set $D_{error}$ based on whether the model’s answer is correct.
It is unclear whether the model was prompted to answer each question **only once**, and what **sampling strategy** was used during token generation. These factors could directly influence the **distribution** differences between the two sets.

- For example, within a single dataset, the model might find questions on some latent topic A easier to answer correctly while questions on topic B are more prone to errors.
In this case, the observed distance differences between the correct and error sets might naturally arise from the distribution differences in $D_{correct}$ and $D_{error}$.
It is unclear how the authors accounted for or mitigated such effects.

- To ensure that the two sets in the dataset have as similar a distribution as possible, my understanding is that, for the same question, the model could sample multiple correct and incorrect reasoning paths. Comparing the reasoning manifolds between these two types of paths might yield more meaningful insights.

2. The observation that hidden states of large language models reside in a low-dimensional space aligns with general understanding but does not provide additional insights.

3. The proposed method is relatively straightforward, and it is unclear how the choice of several hyperparameters might affect the conclusions (e.g., the number of neighbors in Section 3.3, the $\alpha$ threshold in Section 3.4). The authors could provide more explanation or justification for these choices.

4. The method primarily focuses on locating failure cases and analyzing deviations in representations, while there is limited discussion on why the model produces these failures and how such failure could be mitigated or corrected.

**Questions:**

see Weakness.

---

### Official Review · Reviewer_9U9B · 2025-10-30

**Soundness:** 3
**Presentation:** 3
**Contribution:** 2
**Rating:** 4
**Confidence:** 3

**Summary:**

This paper presents REMA, a post hoc interpretability framework designed to analyze and localize reasoning failures in large language models. The work introduces the "Reasoning Manifold" as its core theoretical concept. This concept posits that the internal model representations corresponding to correct reasoning processes are not scattered randomly but form a coherent, low dimensional geometric structure within the high dimensional activation space.

The REMA framework operationalizes this hypothesis by first approximating the manifold using the point cloud of hidden states from correctly answered samples. Reasoning failures are then defined and quantitatively measured as geometric deviations from this "correct" manifold, calculated using a k nearest neighbors distance metric. The framework also seeks to identify the origin of a failure by tracking this deviation layer by layer for a given error sample. This allows for the localization of a "divergence point", which is the earliest layer where the representation's distance from the correct manifold surpasses a statistical threshold. The authors provide empirical validation across several models and tasks, showing that correct and erroneous reasoning representations are geometrically separable and that this separability increases with model depth.

**Strengths:**

The paper's primary contribution is the formalization of LLM reasoning failures as a problem of geometric deviation. The introduction of the "Reasoning Manifold" concept, which posits that correct reasoning paths form a low dimensional structure, provides a theoretically grounded and intuitive anchor for the subsequent analysis. The proposed REMA framework is a practical and well specified implementation of this idea. It provides a clear method for not only quantifying failure severity using k-NN distances but also for localizing the origin of the failure to a specific "divergence point" in the model's layers.

In their empirical validation, the authors use a wide array of models, including both text and multimodal architectures of various scales, and test them across a diverse set of reasoning benchmarks. The results demonstrate that correct and erroneous representations are geometrically separable.

**Weaknesses:**

While the paper introduces an intuitive framework, its primary technical weakness lies in the disconnect between the "Reasoning Manifold" concept and its simple implementation. The framework does not actually model a manifold. Instead, it approximates this complex geometric structure as a simple point cloud of correct sample representations. As a result, the analysis is based on standard Euclidean k nearest neighbor distances. The work would be substantially more rigorous if it engaged with actual manifold learning techniques to estimate local structure, such as tangent spaces, or used manifold-aware distance metrics like geodesic distances. This would provide a far more robust validation of the geometric hypothesis beyond what standard outlier detection in a vector space can offer.

A second, and perhaps more significant, technical limitation is the choice of representation via mean pooling across all generation tokens. A reasoning process is fundamentally a *trajectory* or a sequence of states, but by averaging all hidden states at a layer into a single vector, the framework discards all temporal dynamics. A failure that occurs at a single critical generation step may be "washed out" by this averaging. This choice directly undermines the stated goal of failure localization. The paper identifies a "divergence point" at a specific *layer*, but this is a very coarse measure. Because the mean pooled vector at layer $l$ contains information from all generation steps $t=1...T_i$, it is not a clean indicator of *when* the reasoning process first went off track.

**Questions:**

1. Why a point cloud approximation was chosen for the manifold, rather than more formal manifold learning techniques? For instance, did the authors experiment with estimating local tangent spaces or using an approximated geodesic distance to better align with the paper's central theoretical claim? How robust is the k-NN deviation metric to the sampling density of correct representations? It seems possible that a novel, yet correct, reasoning path could be distant from its neighbors in a sparse manifold and be incorrectly flagged as a deviation.

2, Regarding the mean pooling strategy, how can this single, aggregate vector distinguish a failure that begins at the first generation token from one that occurs at the final token? The localization to a *layer* seems to be an average over the entire generation process. Why is this aggregate state preferable to analyzing the state at each specific generation step? A more powerful localization would seem to require analyzing the deviation of sequential states from a manifold of correct *partial* reasoning paths.

3. The paper shows that the Intrinsic Dimension (ID) of the representations is low, but was this ID estimate actually used in the REMA framework? Projecting representations onto this low dimensional subspace before computing distances might provide a cleaner deviation signal. Also, the choice of the threshold factor $\alpha=2$ seems heuristic and the appendix confirms its high sensitivity. Could the authors provide a more principled justification for this value, or discuss how a practitioner should set this parameter?

---

### Note · Authors · 2026-01-26

I have read and agree with the venue's withdrawal policy on behalf of myself and my co-authors.

---

### Meta-Review · Area_Chair_EFUN · 2026-01-06

**Summary:**

This paper proposes a new framework "REMA" to quantify LLM reasoning failures. The key idea is to formalize LLM reasoning failures as geometric deviations from a low dimension reasoning manifold which represent correct reasoning paths. The major problem with this paper is the fundamental mismatch between its theoretical claim in the paper and the actual implementation/methodology in the paper. There are also other concerns from the reviewers including flawed evaluation, limited experiment details and ablation study. The authors did not respond to the reviewers concerns in the rebuttal period, hence all the concerns remains and a rejection is recommended.

**Reviewer Concerns:**

none of the concerns are addressed, since the authors did not respond in the rebuttal

**Reviewer Scores:**

none of the reviewers will change score since no rebuttal is provided

---

### Decision · Program_Chairs · 2026-01-26

Reject